# Weed Communities in Winter Wheat: Responses to Cropping Systems under Different Climatic Conditions

Tim Seipel [1], Suzanne L. Ishaq [2], Christian Larson [1] and Fabian D. Menalled [1,*]

[1]  Department of Land Resources and Environmental Sciences, Montana State University, Bozeman, MT 59717, USA; timothy.seipel@montana.edu (T.S.); christian.larson@montana.edu (C.L.)
[2]  School of Food and Agriculture, University of Maine, Orono, ME 04469, USA; sue.ishaq@maine.edu
*  Correspondence: menalled@montana.edu; Tel.: +1-406-994-5883

**Abstract:** Understanding the impact of biological and environmental stressors on cropping systems is essential to secure the long-term sustainability of agricultural production in the face of unprecedented climatic conditions. This study evaluated the effect of increased soil temperature and reduced moisture across three contrasting cropping systems: a no-till chemically managed system, a tilled organic system, and an organic system that used grazing to reduce tillage intensity. Results showed that while cropping system characteristics represent a major driver in structuring weed communities, the short-term impact of changes in temperature and moisture conditions appear to be more subtle. Weed community responses to temperature and moisture manipulations differed across variables: while biomass, species richness, and Simpson's diversity estimates were not affected by temperature and moisture conditions, we observed a minor but significant shift in weed community composition. Higher weed biomass was recorded in the grazed/reduced-till organic system compared with the tilled-organic and no-till chemically managed systems. Weed communities in the two organic systems were more diverse than in the no-till conventional system, but an increased abundance in perennial species such as *Cirsium arvense* and *Taraxacum officinale* in the grazed/reduced-till organic system could hinder the adoption of integrated crop-livestock production tactics. Species composition of the no-till conventional weed communities showed low species richness and diversity, and was encompassed in the grazed/reduced-till organic communities. The weed communities of the no-till conventional and grazed/reduced-till organic systems were distinct from the tilled organic community, underscoring the effect that tillage has on the assembly of weed communities. Results highlight the importance of understanding the ecological mechanisms structuring weed communities, and integrating multiple tactics to reduce off-farm inputs while managing weeds.

**Keywords:** conventional agriculture; organic agriculture; reduced tillage; crop-livestock integration; climate change

## 1. Introduction

Despite numerous technological advances and knowledge gain in weed management, weeds continue to challenge agricultural production, especially in the face of global climate change [1–3]. For example, small grain and pulse crop production represents a major economic activity in the Northern Great Plains of the United States [4]. Yet, the intensification of crop production and the heavy reliance on off-farm inputs occurring in this region resulted in unintended consequences, including the selection of a specialized pathogen, insect pest, and weed complex [5] that includes multiple herbicide-resistant biotypes [6]. The difficulty to manage these herbicide-resistant biotypes, coupled with predicted increases in temperatures and a reduction in summer precipitation further threaten the long-term sustainability of semi-arid agroecosystems [7,8].

Cropping systems act as a set of distinct ecological filters that structure the abundance and relative composition of the associated agricultural diversity, including weed

communities [9]. In the Northern Great Plains, conventional row crop production can be characterized by simplified crop-fallow rotations coupled with heavy reliance on off-farm synthetic inputs in the form of pesticides and fertilizers to manage pests and secure yields. In contrast, organic production relies on diversified crop rotations, cover crops, and tillage to manage soil fertility and pests. Previous studies conducted in this region [10–12] determined that the different suites of management practices associated with these systems resulted in distinct weed communities, with an overall increased abundance, species richness, and species diversity, as well as more complex spatial distribution in organic farms than in conventionally managed ones.

Predicted higher temperatures and reduced moisture availability, combined with novel biotic interactions are expected to reduce agricultural production, especially of wheat [13,14]. Warmer and drier climate conditions can reduce agroecosystem resiliency [15], increase crop stress [16], and lead to shift in weed communities [17]. For example, ref. [18] indicated that higher temperatures could result in an increase in the relative abundance of thermophile and late-emerging species. Further, predicted increases in carbon dioxide levels coupled with warmer and drier climates are expected to compromise the efficacy of weed management strategies [3,19].

Mounting concerns about the negative outcomes of predicted climate conditions on agricultural production have spurred interest in the development of ecologically-based cropping systems that include practices such as cover cropping, crop diversification, and diversified production practices [20]. One example is the integration of crop and livestock production where targeted grazing is used to terminate cover crops and manage weeds [21]. Previous research indicated that, in the Northern Great Plains, the integration of crop and livestock production in organic systems allows a reduction in tillage intensity [11] but can impact the structure of weed [11,22], invertebrate [23], and soil microbial [24–26] communities. Yet, to our knowledge, no study has specifically compared the impact of predicted increased temperature and reduced moisture conditions on weed communities among conventional and organic cropping systems with varying degrees of soil disturbance. To assess this knowledge gap, we evaluated how weed biomass and weed communities of winter wheat crops vary in response to temperature and moisture across three contrasting cropping systems: (1) a no-till conventional system manage with chemical inputs, (2) a tilled-organic system, and (3) a grazed/reduced-till organic system where targeted grazing with sheep (*Ovis aries*) was used to terminate cover crops and manage weeds during pre-seeding and post-harvest with the goal reducing mechanical soil disturbance. We selected winter wheat because of its agricultural importance across the mesic sections of the Northern Great Plains, a semi-arid region encompassing approximately 144 million hectares in central North America. In this region, more than to 7 million ha of hard red winter wheat were grown in 2017 alone [20].

The three studied systems respond differently to changes in temperature and moisture [15] and use different farm inputs and management tactics that can act as distinct ecological filters affecting the composition of weed communities [27]. Thus, we hypothesized a shift in weed abundance and community structure across the studied systems. Specifically, because a reduction in tillage intensity facilitates the establishment of perennial species which are difficult to be managed in organic production [28], we expected higher abundance of this life form in the grazed/reduced-till organic system. Finally, based on previous studies [3,17], we expected that weed community responses to alterations of soil temperature and moisture conditions would differ across the conventional no-till, tilled organic, and grazed/reduced-till organic systems.

## 2. Materials and Methods

### 2.1. Study Site and Experimental Design

Our study was conducted during the 2015–2016 and 2016–2017 growing seasons within a five-year cropping system experiment at the Montana State University's Fort Ellis Research and Extension Center, located east of Bozeman, Mt, USA (45.6671 latitude,

longitude –110.9977, elevation 1500 m a.s.l.) The 30-year mean annual precipitation for Fort Ellis is 518 mm, and the mean annual maximum and minimum temperatures are 13.6 °C and 0.9 °C, respectively. During the experiment, average monthly temperature in the spring growing period (March through June) in both years was slightly above average (0.5 to 1.5 °C), compared with 30 year mean monthly temperatures from 1981–2010 [29]. Precipitation from March to June 2016 was 80% of the 30-year average. During the same period of 2017, precipitation was 110% of the 30-year average (Table S1). Soils at the Fort Ellis site are classified as a silt loam (a fine-silty, mixed, superactive, frigid Typic Arguistoll) with 0% to 4% slopes and consistent ratio of 1 part sand, 2 parts silt, and 1 part clay by weight.

Prior to 2004, the study site was planted with perennial grasses (*Bromus inermis* L., *Thinopyrum intermedium* (Host) Barkworth and D.R. Dewey, and *Poa compressa* L.). From 2004 to 2009, the site was seeded to a continuous spring wheat, a spring wheat-fallow, or a winter wheat-fallow crop rotation. Between 2009 and 2012, the study site was seeded to either a continuous alfalfa (*Medicago sativa* L.), or a three-year crop rotation consisting of spring wheat in the first year followed by pea (*Pisum sativum* L.), and hay barley (*Hordeum vulgare* L.) in the second and third years, respectively. To homogenize potential variability due to these previous activities, the entire site was planted with glyphosate-tolerant rapeseed (*Brassica napus* L.) and treated with herbicides in the spring of 2012. The rapeseed was tilled to a depth of 15 cm in July 2012 and planted in September 2012 following the experimental design described below.

The main cropping system experiment had three replicated blocks. Within each replicate, 75 m by 90 m areas were randomly assigned to one of three management systems: (1) a no-till chemically managed system that relied on synthetic inputs in the form of fertilizers, herbicides, and fungicides to manage nutrient availability, weed abundance, and pathogen pressure, (2) a tilled-organic system reliant on tillage to manage weeds and terminate cover crops, and (3) a grazed/reduced-till organic system that incorporated sheep grazing to manage weeds and terminate cover crops with the overall goal of reducing the tillage intensity. The three cropping systems followed the same five-year rotation with all phases present every year. Crops were seeded in 13 m × 90 m sections, separated by a 1 m fallow strip. Crop phases of the rotation were randomly assigned in 2012 to each section and were: safflower (*Carthamus tinctorius*) undersown with the biennial sweet clover (*Melilotus officinalis*) (year 1), the sweet clover was then grown as a cover crop in the next phase (year 2), followed by winter wheat (*Triticum aestivum* cv. Yellowstone) (year 3), lentil (*Lens culinaris*) (year 4), and winter wheat (year 5). During the first year of the study, it was not possible to have the biennial sweet clover and pea (*Pisum sativum*) was planted instead of sweet clover and terminated as a cover crop. Inputs in the no-till system included 2,4D, bromoxynil, dicamba, fluroxypyr, glyphosate, MCPA, pinoxaden, and urea, which are reflective of typical no-till conventional farm management practices in the Northern Great Plains region. Both organic systems began the organic transition process in July 2012, so that crops harvested after 2015 met USDA organic certification standards. In the organic tilled system, tillage was accomplished using a chisel plow, tandem disk, or field cultivator, as needed for weed control, seedbed preparation, and to incorporate cover crops and crop residues. Weed control was enhanced with a rotary harrow. In the organic grazed/reduced-till system, sheep grazing was used to reduce tillage intensity for pre-seeding and post-harvest weed control, and to terminate the cover crops, with duration and intensity of grazing based on weed pressure or cover crop biomass. During the first three years of the rotation, tillage was eliminated in the organic grazed/reduced-till system, and it was significantly reduced in the other two years of the study. Additional details on the site history, inputs, and maintenance can be found in Table S2 and refs. [11,28].

The current study was carried out as a mini-plot experiment within the year 3 winter wheat crop phase of all the three cropping systems. In both years of the study, we established three 0.75 m$^2$ circular split-plots and randomly assigned them to one of three temperature and moisture treatments: (1) ambient temperature and moisture, used as

control, (2) warmer treatment that increased air and soil temperature using open-top chambers, and (3) a warmer and drier treatment that increased temperature and decreased rainfall using a combination of rainout shelters and open-top chambers (Figure S1) [30]. Open-top chambers followed ref. [31] and were constructed of 1 mm-thick Sun-Lite HP (Solar Components Corporation, Bow, NH, USA) with a basal diameter of 1.6 m and the top opening diameter was 1.0 m. The height of the chambers was 0.5 m and chamber wall had an incline of 65°. Rainout shelters were used to reduce the amount of moisture by approximately 50% [32] and were constructed with a wooden frame and corrugated polycarbonate plastic that covered approximately 50% of a 2 m-by-2 m area centered over the open-top chambers. The rainout shelter was oriented west to east; the west side was lower and faced into the prevalent wind, and the incline of the rainout shelter increased from west to east by approximately 30°. These temperature and moisture manipulation structures were placed in the field as winter wheat emerged from dormancy in early spring (early March 2016 and early April 2017) and were removed after of harvest of wheat in early August. To monitor the impact temperature and moisture manipulations, we recorded soil moisture using Delmhorst gypsum block sensors (https://www.delmhorst.com; accessed on 4 April 2022) and monitored soil temperature 5 cm below the surface using Maxim ibuttons (https://www.maximintegrated.com/; accessed on 4 April 2022).

To assess changes in weed communities as a function of cropping system, and temperature and moisture manipulation, we destructively sampled all the aboveground weed biomass within the 0.75 m² split-plots in mid-July. We cut all weeds at ground level as they began to senesce and no longer had a competitive impact on the ripening wheat. All sampled weeds were separated by species, oven dried until constant weight, and weighed.

### 2.2. Data Analysis

Soil moisture, measured as electrical conductivity, was converted to soil water potential expressed in bars based on [33]. Daily mean temperature was calculated as the average of temperature measured every 3 h by the ibuttons. General additive models (GAM) were used to assess how open-top chambers and rainout shelters affected soil temperature and soil moisture conditions throughout the growing season with the 'mgcv' package in the R statistical environment [34]. The GAM models were fit using day as the predictor, smoothed using a penalized cubic regression spline, and temperature and moisture manipulations as factor levels. Differences in soil temperature and soil moisture across the three studied cropping systems were evaluated using analysis of variance (ANOVA).

To assess differences in total weed biomass across cropping systems, temperature and moisture manipulations, and year of study, we fit linear mixed-effects models using the 'lmerTest' package in R [35]. Total biomass was summed for all species within each temperature and moisture manipulation treatment, and log transformed to meet assumptions and improve model fit. Total weed biomass in response to cropping system, temperature and moisture manipulation treatments, year, and their interactions was fitted using the split-plot as a random effect. Post-hoc pairwise Tukey comparisons of means were conducted using the 'emmeans' package [36] and figures were produced using the 'ggplot2' package [37]. The impact of cropping system, temperature and moisture conditions, and year of study on weed communities was assessed by calculating species richness and the inverse Simpson's diversity metric for each treatment combination. Simpson's diversity was calculated using biomass from each weed species. These variables were compared using generalized linear mixed-effects models with cropping system, moisture and temperature manipulation, year, and their interactions as fixed effect variables, using the split-plot as a random effect. Post-hoc Tukey comparisons among means were conducted when factors accounted for significant portions of variation.

A permutational multivariate ANOVA (PERMANOVA) and algorithm 'adonis' in the 'vegan' package were used to assess if a significant proportion of the variation in individual weed species biomass was accounted by cropping system, temperature and moisture manipulations, and year [38]. In the PERMANOVA models, cropping system,

temperature and moisture manipulations, and their interactions were used as predictor variables. Year was used to constrain variation within trials of the experiment. Principal coordinate analysis and bi-plots were used to visualize differences in dissimilarity among predictor variables based on the results of PERMANOVA.

## 3. Results

### 3.1. Temperature and Moisture Manipulations

Including soil temperature and moisture manipulation treatments better characterized variation in soil temperature and soil moisture than day and year alone ($F_{(20, 33445)} = 10.8$, $p < 0.001$ and $F_{(19.7, 2957)} = 6.8$, $p < 0.001$, respectively). Open-top chambers and the combination of open-top chambers and rain-out shelters increased soil temperature compared to ambient conditions, particularly during the critical spring growing season when wheat tillers, heads, and kernels develop (Figure S2). Soil moisture was reduced in the warmer and drier conditions, especially in 2016 (Figure S3). In 2017, soil moisture in the warmer and drier treatment was lower than in the ambient and warmer treatments during the early April to late June period. However, precipitation above the long-term average (Table S1) resulted in a smaller intra-year difference and higher available moisture compared with 2016.

### 3.2. Weed Biomass and Community Responses

Weed biomass varied among cropping systems and years but was not affected by manipulated climate conditions (Table 1). In 2016, total weed biomass in the no-till conventional system was lower than in both tilled organic ($p < 0.001$) and the grazed/reduced-till organic ($p < 0.001$) systems, while there was no difference between the tilled and grazed organic systems ($p = 0.20$, Figure 1a). Similarly, in 2017 total weed biomass in the no-till conventional system was lower than in both the tilled ($p < 0.001$) and grazed/reduced-till organic ($p < 0.001$) system, and no difference was observed between the tilled and grazed/reduced-till organic systems ($p = 0.55$, Figure 1b). While the trends were the same across years, weed biomass was one order of magnitude greater in 2017 than in 2016 in the tilled and grazed/reduced-till organic cropping systems. Within each cropping system, changes in soil temperature and moisture did not result in differences in weed biomass, but overall variability was higher in the two organic systems than in the no-till conventional systems (Figure 1).

**Table 1.** Type III Analysis of Variance of aboveground dry weed biomass in response cropping system, temperature and moisture manipulations, year of the experiment, and their interactions. Satterthwaite approximation was used to estimate degrees of freedom.

| | Sum of Squares | Mean Square | Numerator Degrees of Freedom | Denominator Degrees of Freedom | F Value | p Value |
|---|---|---|---|---|---|---|
| Cropping system | 1021.6 | 510.79 | 2 | 85.02 | 139.0 | <0.001 |
| Temperature and moisture manipulation (T&M) | 5.47 | 2.74 | 2 | 85.02 | 0.7 | 0.478 |
| Year | 193.51 | 193.51 | 1 | 85.02 | 52.7 | <0.001 |
| Cropping system × T&M | 29.66 | 7.41 | 4 | 85.02 | 2.0 | 0.099 |
| Cropping system × Year | 35.98 | 17.99 | 2 | 85.02 | 4.9 | 0.010 |
| T&M × Year | 4.68 | 2.34 | 2 | 85.02 | 0.6 | 0.532 |
| Cropping system × T&M × Year | 7.31 | 1.83 | 4 | 85.02 | 0.5 | 0.738 |

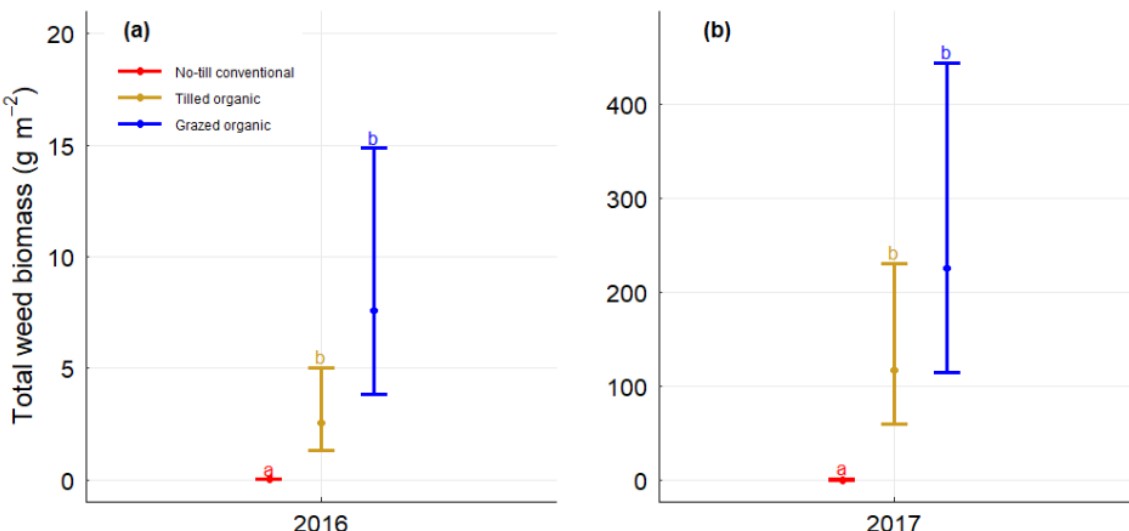

**Figure 1.** Estimated means of total weed biomass (g m$^{-2}$) in 2016 (**a**), and 2017 (**b**) by cropping system. Means are back transformed estimated marginal means derived from the response variable in the linear mixed-effects model. Error bars indicate SE of the mean. Note the difference in scales between years. Letters indicate differences among means ($p \le 0.05$).

Weed species richness varied among cropping systems ($p = 0.007$), but not in response to the year of the trial ($p = 0.80$). Soil temperature and moisture manipulations did not affect weed species richness ($p = 0.94$), nor interact with cropping systems ($p = 0.41$). A total of 32 weed species were sampled across all treatments and years (Table 2). The largest number of weed species was observed in the grazed/reduced-till organic cropping system, which had a total of 20 species and an average of 4.8 species ($\pm 1.2$ SE) per 0.75 m$^2$ split-plot (Figure 2). A total of 11 species were recorded in the tilled organic system, with an average of 2.2 species ($\pm 1.2$ SE) per 0.75 m$^2$ split-plot per year. The no-till conventional cropping system had the lowest number of weed species (three, total) and the lowest average of number of weed species per 0.75 m$^2$ split-plot (1.1 $\pm$ 0.2 SE). A detail analysis of the impact of cropping system on weed communities under current temperature and moisture conditions can be found in ref. [11].

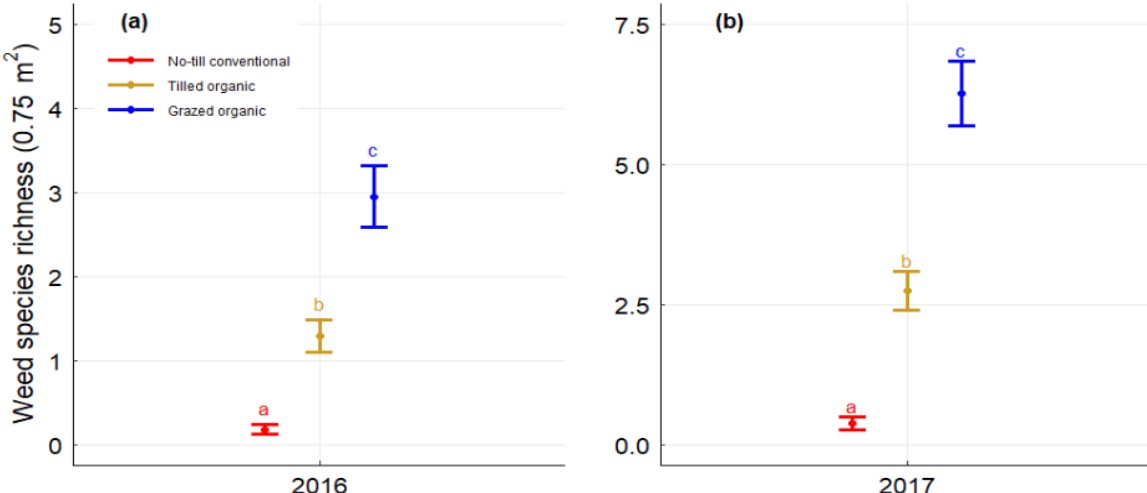

**Figure 2.** Mean weed species richness in 0.75 m$^2$ split plots for (**a**) 2016 and (**b**) 2017 across the no-till conventional, tilled organic, and grazed/reduced-till organic cropping systems. Error bars indicate SE of the mean. Letters indicate differences among means ($p \le 0.05$).

**Table 2.** Total dry biomass of weed species sampled in the three cropping systems summed across the two years of the study (total area sampled per treatment = 13.5 m$^2$).

| No-Till Conventional | | Grazed/Reduced-Till Organic | | Tilled Organic | |
|---|---|---|---|---|---|
| **Weed Species** | **Total Dry Biomass (g)** | **Weed Species** | **Total Dry Biomass (g)** | **Weed Species** | **Total Dry Biomass (g)** |
| *Bromus tectorum* | 17.83 | *Bromus tectorum* | 1429.8 | *Thlaspi arvense* | 2221.3 |
| *Lactuca serriola* | 16.86 | *Sisymbrium altissimum* | 1296.0 | *Lactuca serriola* | 615.3 |
| *Chenopodium album* | 0.02 | *Thlaspi arvense* | 1092.0 | *Asperugo procumbens* | 87.7 |
| | | *Lactuca serriola* | 622.7 | *Capsella bursa-pastoris* | 27.2 |
| | | *Asperugo procumbens* | 476.1 | *Cirsium arvense* | 22.4 |
| | | *Taraxacum officinale* | 196.6 | *Camelina microcarpa* | 10.3 |
| | | *Capsella bursa-pastoris* | 150.7 | *Sisymbrium altissimum* | 7.4 |
| | | *Tragopogon dubius* | 137.5 | *Tragopogon dubius* | 3.5 |
| | | *Melilotus officinalis* | 89.0 | *Chenopodium album* | 1.1 |
| | | *Cirsium arvense* | 71.0 | *Taraxacum officinale* | 0.9 |
| | | *Trifolium pratense* | 32.9 | *Bromus tectorum* | 0.6 |
| | | *Bromus japonicus* | 15.8 | | |
| | | *Xanthium strumarium* | 7.2 | | |
| | | *Dactylis glomerata* | 7.0 | | |
| | | *Descurainia sophia* | 6.5 | | |
| | | *Agropyron cristatum* | 4.7 | | |
| | | *Galium aparine* | 3.8 | | |
| | | *Lotus corniculatus* | 3.0 | | |
| | | *Medicago sativa* | 0.6 | | |
| | | *Avena fatua* | 0.5 | | |

Inverse Simpson's diversity index varied among cropping systems ($p \leq 0.001$), between years ($p = 0.006$), and was greatest in the grazed/reduced-till organic cropping system ($p = 0.09$), with no significant interactions among variables. Weed community diversity, as estimated by the Inverse Simpson index, was not impacted by soil temperature and moisture manipulations ($p = 0.32$), or its interaction with cropping systems ($p = 0.36$). In both years, the tilled organic and no-till conventional cropping systems were the least diverse and showed similar levels of weed species diversity, as estimated by the Inverse Simpson's index ($p = 0.17$). The grazed/reduced-till organic system was more diverse than the other two cropping systems (Figure 3a,b).

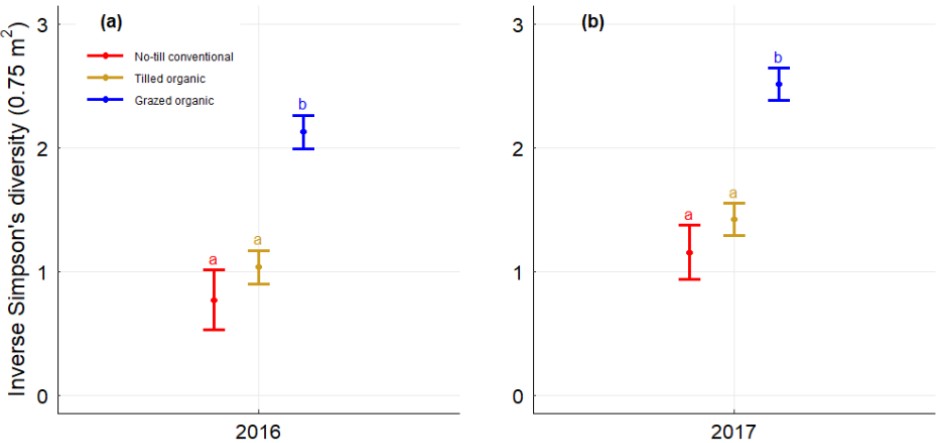

**Figure 3.** Mean inverse Simpson's diversity index in 0.75 m$^2$ split plots for 2016 (**a**) and (**b**) 2017 (**b**) across the no-till conventional, tilled organic, and grazed/reduced-till organic cropping systems. Error bars indicate SE of the mean. Letters indicate differences among means ($p \leq 0.05$).

Weed community composition varied in response to cropping system ($p = 0.001$), which accounted for 16% of total variation (Table 3, Figure 4). While significant ($p = 0.041$), temperature and moisture manipulations accounted for only 3% of total variation in weed communities (Table 3, Figure 4). The tilled organic and grazed/reduced-till organic cropping systems were more variable in the ordinal space than the no-till conventional system, reflecting their greater variability in biomass (Figure 1 and see the ellipses in Figure 4). The no-till conventional system had the least variation in composition among the systems (Figure 1 and, see the ellipses in Figure 4) and was nested within one standard deviation of grazed/reduced-till organic system (Figure 4).

**Table 3.** PERMANOVA of weed species composition in response to temperature and moisture manipulations and cropping system based on dissimilarity of species biomass using the Bray–Curtis index. Year was as the constrain variable.

| | Degrees of Freedom | Sums of Squares | Mean Squares | F-Value | $R^2$ | $p$ |
|---|---|---|---|---|---|---|
| Temperature and moisture manipulation (T&M) | 2 | 0.9 | 0.4 | 1.3 | 0.03 | 0.041 |
| Cropping system | 2 | 4.6 | 2.3 | 6.9 | 0.16 | 0.001 |
| Cropping system × T&M | 4 | 1.4 | 0.4 | 1.0 | 0.05 | 0.105 |
| Residuals | 68 | 22.9 | 0.3 | | 0.77 | |

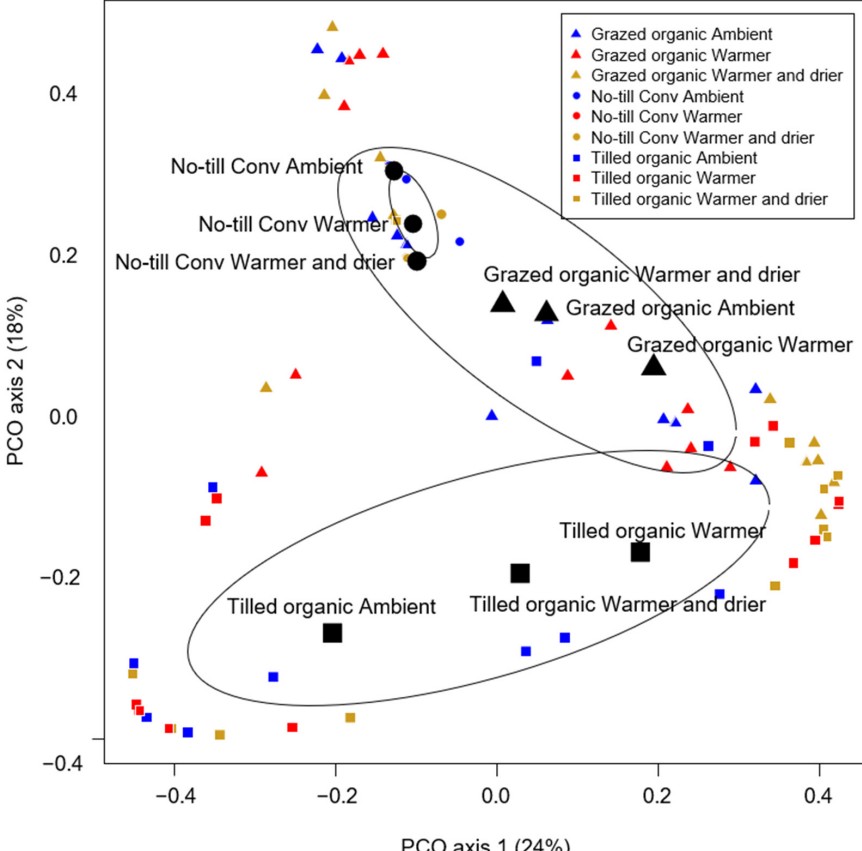

**Figure 4.** Principal coordinates ordination bi-plot of the dissimilarity of weed species biomass grouped by cropping system and temperature and moisture manipulations. Ellipses indicate the standard deviation of centroids of cropping systems. Black symbols indicate the centroids of temperature and moisture manipulations within each cropping system. Dissimilarity was calculated using the Bray–Curtis index based on biomass of weed species in each split-plot.

Across all cropping systems, winter annuals were the most common life cycle (Table 2), and included *Bromus tectorum*, *Sisymbrium altissimum*, and *Lactuca serriola*. The relative abundance of species differed across cropping systems. Specifically, *Thlaspi arvense* L dominated the weed communities sampled in the tilled organic system (Table 3) and *B. tectorum* was the most abundant weed species in the grazed/reduced-till organic and no-till conventional system. While two perennial weed species (*Cirsium arvense* and *Taraxacum officinale*) were abundant in the grazed/reduced-till organic systems, less than 1 g of *T. officinale* was sampled in the tilled organic system, and no perennial species were found in the no-till conventional system (Table 2).

## 4. Discussion

Human population growth, coupled with shifts in consumer and market demands, new societal concerns about environmental impacts of agricultural practices, and unprecedented climatic conditions makes it necessary to understand how cropping systems respond to biological and environmental stressors [39]. In this context, there is a need to develop tactics with reduced reliance on external inputs to manage pests and secure yields [40,41]. Ecologically based tactics such as the integration of crop–livestock production and the adoption of reduced input systems, including organic practices, have been proposed as approaches to achieve these goals [21,40]. In this paper, we assessed how predicted warmer and drier conditions [7] could impact weed communities in winter wheat across three contrasting cropping systems: a no-till chemically managed conventional system and two organic systems, one reliant on tillage and another where targeted grazing with sheep was used to terminate cover crops and manage weeds with the goal reducing mechanical soil disturbance. Winter wheat dominates crop production in the dryland sections of the Northern Great Plains [20], but higher temperatures are expected to affect production [13]. For example, a 1 °C increase in global temperature is estimated to reduce wheat yield by 8% [42].

In agroecosystems, management tactics represent distinct ecological filters of plant communities, selectively favoring some taxa while excluding others [9,43]. The results of our research confirmed that cropping systems have an overriding impact on weed community characteristics. In agreement with our first expectation and previous studies conducted in this region [10,12], more weed biomass, and increased species richness and diversity were observed in the tilled organic system compared with the no-till conventional cropping system. In agreement with our second expectations, it was also observed that there was an increased abundance in difficult-to-manage perennial weed species [44] in the grazed/reduced-till organic system, a shift that could hinder the adoption of integrated crop and livestock production tactics. In partial contradiction with our last expectation, weed community responses to temperature and moisture manipulations differed across variables: while biomass, species richness, and Simpson's diversity estimates did not differ across treatments, a significant, but minor shift, in weed community composition was observed as a function of temperature and climate manipulations.

The results of this study concur with previous research indicating that tillage represents a unique soil disturbance affecting the structure and dynamics of weed communities [43,45]. *Thlaspi arvense*, a winter annual species commonly found in organic small grain cropping systems of the Northern Great Plains [10], was the most common species sampled in the tilled organic system but was absent in the no-till conventional system and a minor component of the grazed/reduced-till organic system. In contrast, *B. tectorum*, a winter annual species that is known to flourish in reduced-tillage systems [46] dominated the weed communities of the no-till conventional and the grazed/reduced-till organic systems. The increased abundance of *B. tectorum* highlights future management challenges as habitat suitability of this species is expected to surge under predicted warmer and drier conditions [47]. Interestingly, the weed community in the no-till conventional system had lower diversity and biomass, was encompassed within the ordination space of the grazed/reduced-till organic community, and these two weed communities were different from those sampled

in tilled organic cropping system. The similar weed community observed between the no-till conventional and the grazed/reduced-till organic systems could be a result of the absence of tillage in these two systems for 36 months prior to this study [28], underscoring the effect that soil disturbance has on the assembly weed communities.

When comparing weed communities of organic cropping systems with different tillage intensity, ref. [45] observed that *C. arvense* dominated the systems with reduced-tillage intensity. In agreement, an increase *C. arvense* and *T. officinale* abundance was observed in the grazed/reduced-till organic system compared with the tilled organic system. Perennial weed species are a particular management challenge in organic cropping systems [44,48], and the increase in *C. arvense* and *T. officinale* biomass observed in the grazed/reduced-till organic system indicates that grazing may not a reliable method as a unique control to deal with problematic weed species. While a meta-analysis revealed that integrated weed management practices appear to be the most promising approach to manage perennial weeds in organic cropping systems [44], our results highlight the inadequacy of integrating crop and livestock operations in organic systems to reduce tillage intensity while minimizing perennial weeds.

In a recent study, ref. [17] observed that in a semi-arid and cold section of the Northern Great Plains, an increase of summer temperatures coupled with a reduction in moisture availability differentially impacted the performance of early-season and mid-season cover crop mixtures. These shifts in cover crop performance and weed community composition associated with a decrease in weed species richness and diversity, and a shift in weed community composition. This study was conducted in a relatively wetter section of the Northern Great Plains In contrast to ref. [17], and we observed only a minor shift in weed communities across cropping systems when they were exposed to predicted warmer and drier conditions. It is possible that the relatively short-term nature of the study, the relatively high moisture availability at the study site, and the large range of temperature and precipitation that occurred during two years of our study reduced our ability to detect the existence of any impact of projected climate conditions on weed communities. Additionally, the relatively small size of the open top chambers and rainout shelters has been cited as a shortcoming of in situ climate manipulations [49], and could have masked our ability to detect the impacts of predicted warmer and drier climate conditions on weed communities across contrasting cropping systems. Yet, results underscore the importance that management decisions coupled with abiotic and biotic constraints have in determining the structure of agricultural weed communities [9].

## 5. Conclusions

In the semi-arid sections of the Northern Great Plains of the United States, a predicted increase of summer temperatures coupled with a reduction in moisture availability could impact weed communities [17] and the efficacy of weed management strategies [3]. To the best of our knowledge, this is the first evaluation of the impact of predicted climate conditions across a range of conventional and organic cropping systems. This study showed that while cropping system characteristics represent a major driver structuring weed communities, short-term impact of changes in temperature and moisture conditions appear to be more subtle. The no-till chemically managed and organic-tilled systems had the lowest levels of weed abundance, but they also associated with the lowest levels of biodiversity. The extent to which these differences translate in crop yield and other ecosystem services such as pollination, soil erosion, soil health, and resource availability are beyond the scope of this study. Nevertheless, these results suggest that the potential benefits of crop and livestock integration may come with the cost of increased weed pressure, particularly of perennial species. These observations highlight the importance of understanding the mechanisms underpinning the assembly of weed communities under current and predicted climate scenarios to integrate ecologically based processes in the design of sustainable cropping systems.

**Supplementary Materials:** The following supporting information can be downloaded at: https://www.mdpi.com/article/10.3390/su14116880/s1. Figure S1: Photos of the study with temperature (open top chambers) and moisture manipulation (rainout shelters) structures; Figure S2: Daily soil temperature 5 cm below the surface in winter wheat fields with ambient conditions and climate manipulations that made the soil temperature warmer using open-top chambers, and warmer and drier using open-top chambers and rainout shelters that block 50% of the area above the plots; Figure S3: Seasonal variation in soil moisture 5 cm below the surface in winter wheat fields with ambient conditions and climate manipulations that made the soil temperature warmer using open-top chambers, and warmer and drier using open-top chambers and rainout shelters that block 50% of the area above the plots; Table S1: Average monthly temperature and total monthly precipitation from PRISM during important winter wheat growth stages in 2016 and 2017, when the experiment was conducted, and the 30-year average from 1981–2010; Table S2: Agronomic management details in cover crops and winter wheat across conventional no-till, reduced-till grazed organic, and tilled organic between 2013 to 2017 at Fort Ellis Research and Extension Center in Bozeman, MT, USA.

**Author Contributions:** Conceptualization, F.D.M.; methodology and data analysis, T.S., S.L.I., C.L. and F.D.M.; writing—original draft preparation, T.S., S.L.I. and F.D.M.; writing—review and editing, F.D.M.; funding acquisition, F.D.M. All authors have read and agreed to the published version of the manuscript.

**Funding:** This study was funded by a grant from USDA-ORG grant 2015-51106-23970 and USDA-Agriculture and Food Research Initiative—Foundational and Applied Science Program grant 1014774.

**Institutional Review Board Statement:** Not applicable.

**Informed Consent Statement:** Not applicable.

**Data Availability Statement:** Data available upon request.

**Acknowledgments:** We would like to thank Perry Miller, Jeff Holmes, and Devon Ragen for their help in managing the research site.

**Conflicts of Interest:** The authors declare no conflict of interest.

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
