# Peer review of "Weed Communities in Winter Wheat: Responses to Cropping Systems under Different Climatic Conditions"

_sustainability, doi:10.3390/su14116880_

Round 1

Reviewer 1 Report

Comments on “Weed communities in winter wheat: responses to cropping systems and predicted warmer and drier climate conditions”:

The article by Seipel et al. reported the effect of simulated climate change factors on weed biomass and community under different cropping systems. They found the grazed/reduced till organic system had more weed biomass than others. Overall, the topic was interesting, the experiments were properly designed, the analyses were sound, and the manuscript was well organized and presented. I only have several minor questions upon the acceptance of this work.

Materials and Methods:

L110: replace “non-till” with “no-till”.

L182: perMANOVA or PERMANOVA? Please be consistent.

Results:

L213: “average <1 g” was not precise enough, can you provide more specific data?

L224: Multiple comparisons among different cropping systems may be more helpful.

L227-229: the representation format of X2 and its P value should be consistent throughout the article

Discussion:

L316 and 326: check the citation format

Author Response

Reviewer 1.

  • L110: replace “non-till” with “no-till”.

Modified as suggested.

  • L182: perMANOVA or PERMANOVA? Please be consistent. 

Modified to PERMANOVA

  • L213: “average <1 g” was not precise enough, can you provide more specific data? 

Thank you for the observation, we have changed the data within the text for more precision.

  • L224: Multiple comparisons among different cropping systems may be more helpful. 

We wish to focus our attention on the most important factors determining the structure of weed communities. In Table 1, the temperature and moisture treatment were only marginally significant (P=0.099), and the cropping system treatment by far explained the most variation (Table 1).  Thus, we conducted multi-comparisons at the cropping system level.

  • L227-229: the representation format of X2 and its P value should be consistent throughout the article

Modified as suggested.

  • L316 and 326: check the citation format  We are not sure what the problem with the references is. 

We have revised the references and included new citations.  

Reviewer 2 Report

The paper discusses contemporary responses of winter wheat weedy communities to cropping systems and anticipated warmer and drier climate conditions. Given the small number of papers written on this topic, the manuscript is noteworthy. Topics include a new approach, in accordance with organic production and herbicides avoidance in weed control. Contemporary and original topic, well-conceived and written manuscript with the possibility of applying the results. The authors followed the journal instructions  for writing. 

Topics include a new approach, in accordance with organic production and herbicides avoidance in weed control. 

The only remark is that in the whole text, writing in the third person singular should be avoided.

Author Response

Reviewer 2

  • Reviewer 2 indicated “The only remark is that in the whole text, writing in the third person singular should be avoided.” However, across the manuscript Reviewer 2 inserted many comments to avoid the first person (i.e., “avoid we” “the third person singular ... should be applied in whole text”).

We interpret Reviewer 2 suggestion that we should use the third singular person.  Accordingly, we modified the whole text and use the third person.

Reviewer 3 Report

The study has been carried out to understand how cropping systems, elevated temperature and moisture stress interactively influence weed communities in Winter wheat. This study is unique as few studies were conducted to understand how climatic changes and cropping systems interactively determine weed communities. Therefore, the outcome of this study will improve our understanding of weeds and how they respond to crop management and climate change. The experimental design and most of the statistical analysis is appropriately conducted in this study. Therefore, the results have validity. However, there are a lot of areas that this manuscript needs to improve before publication. I have made some general points below and made more comments within the manuscript.

Introduction

Introduction better needs to be strengthened with more literature on how different cropping systems (organic vs. conventional) can cause the difference in weed communities. Also, more literature is required on how climatic conditions can shape weed communities. Since tillage is a major factor, more literature is needed on this. A strong hypothesis will benefit what you expect with the cropping systems by climate interactions.

Methodology

The methodology seems to be complicated to understand. Authors need to simplify the details using a table to illustrate the crop rotations and the practices. Mainly this study is conducted in the winter wheat phase and thus requires more specific information about winter wheat management, including weed control practices.

Results

Results and discussion require a lot of organization. It is difficult to follow some of the results explained. Proper use of the ANOVA table can help the reader to understand the interactions than when they were just randomly explained in the text. Need to follow the significant interactions for comparing treatment means. Ex- Figure 1 does not follow the results from the ANOVA table in Table 1.

Some of the important results are in the supplementary figures or tables. It is better to show them in the main text. According to the objectives, identifying how communities respond to cropping systems and climate is key. Therefore, the results should focus on how the major species identified are associated with treatments. Instead, the results touch upon the surface of variation and do not provide detail. Ex- The species associated with different treatments were not depicted in the ordination diagram.

Statistical analysis

When modeling the climate treatment and actual soil temp and moisture, it should have been better to incorporate the cropping systems effect into the same model. Then interactions would have been identified on soil temp and moisture.

Some sections were complicated to understand. The authors mention a generalized linear mixed model was used and, at the same time, say the data were log-transformed. If generalized models were used no need for data transformation.

Discussion

It is not properly structured and organized. Start with the main interactions observed/not observed and discuss around that. Since the climate is a major factor under control in this study, start the discussion about the effect of climate. The discussion needs to be strengthened with further discussion of observations. You discuss species shift, but these data were shown in supplementary tables, not in the main text.

Conclusion- Needs to focus on what you found and what not.

Abstract

This section is pretty well written and more concise.

Author Response

Reviewer 3

  • I have made some general points below and made more comments within the manuscript.

All suggestions made by Reviewer 3 within the manuscript have been accepted and the text has been modified accordingly.

  • Introduction: needs to be strengthened with more literature on how different cropping systems (organic vs. conventional) can cause the difference in weed communities. 

We have significantly modified the Introduction section of our manuscript.  As suggested by Reviewer 3, we added a new paragraph addressing the agronomic and ecological differences of conventional and organic agriculture.  Also, added in this new paragraph and across the Introduction section, we have several references. 

  • Also, more literature is required on how climatic conditions can shape weed communities. Since tillage is a major factor, more literature is needed on this.

We added a sentence and a reference indicating that an increase in temperature could result in an increase in the relative abundance of thermophile and late-emerging species. Also, The reviewed manuscript discuses the impact that predicted climate and CO2 scenarios could have on the efficiency of weed management tactics.

  • A strong hypothesis will benefit what you expect with the cropping systems by climate interactions. 

We modified the last paragraph of the Introduction section to state our hypothesis.  We have also presented our expectations.

  • The methodology seems to be complicated to understand. Authors need to simplify the details using a table to illustrate the crop rotations and the practices. Mainly this study is conducted in the winter wheat phase and thus requires more specific information about winter wheat management, including weed control practices.

To facilitate the interpretation experimental design and field methods, we have included additional information on the agronomic practices and crop rotation used in this study. Specifically, we have provided two additional references with information related to agronomics inputs and management tactics used in each cropping system. Also, we included a new supplemental table (Table S2.) where we provide detailed information of the agronomic management practices employed in the cover crop and winter wheat phases across conventional no-till, reduced-till/ grazed organic, and tilled organic between 2013 to 2017. While this study was conducted in the year 3 (winter wheat phase) of the crops rotation we have included detailed information related the cover crop phase to reflect the grazing operation activities in the reduced till/grazed organic system (as requested by Reviewer 3 in another comment.)

  • Results and discussion require a lot of organization. It is difficult to follow some of the results explained. Proper use of the ANOVA table can help the reader to understand the interactions than when they were just randomly explained in the text. Need to follow the significant interactions for comparing treatment means. Ex- Figure 1 does not follow the results from the ANOVA table in Table 1.  

We have made several changes to improve our Results and Discussion sections.  For example, we removed a paragraph of section 3.2 (Weed biomass and community responses) to ease the interpretation of our Results. We have altered Figures 1 and 2 to show the interactions of cropping system and year to more accurately depict the significant relationships found in Table 1. We have re-worked the richness and diversity paragraphs and added another graph to help clarify our results.  Also, we have modified the Discussion section (see changes in the manuscript and comments below)

  • Some of the important results are in the supplementary figures or tables. It is better to show them in the main text. According to the objectives, identifying how communities respond to cropping systems and climate is key. Therefore, the results should focus on how the major species identified are associated with treatments. Instead, the results touch upon the surface of variation and do not provide detail. Ex- The species associated with different treatments were not depicted in the ordination diagram.

As explained in the Results section of the revised version of our manuscript, we found minimal indication of a consistent shift in weed communities as a function on temperature and moisture conditions.  Because the differences among weed communities are presented for ambient condition in Larson et al., 2021 (see the literature cited) and we direct the reader to this reference for more information.  Still, we provide the relative abundance of weed species among cropping systems in Table S3.

  • When modeling the climate treatment and actual soil temp and moisture, it should have been better to incorporate the cropping systems effect into the same model. Then interactions would have been identified on soil temp and moisture.

We assessed this suggestion and found that cropping systems did not have a discernable impact that we could be easily assessed given degrees of freedom available in this study. For example, the figure below includes cropping system and temperature and moisture manipulation, but add no significant information to Figure S2. 

  • Some sections were complicated to understand. The authors mention a generalized linear mixed model was used and, at the same time, say the data were log-transformed. If generalized models were used no need for data transformation.

Reviewer 3 is correct. We needed to log transform the biomass data but fit only a linear mixed effects model that used the gaussian distribution and then presented the back transformed estimated marginal means. We have clarified this in the revised version of our manuscript.

  • Lines 163-165: “You could have used cropping systems s a predictor in this model and test the interactions with climatic treatments instead of using a separate ANOVA mention below”

Please see our response above.

  • “line 163: “using GAM function?”  

We have clarified that we used general additive models to assess how the climate manipulations affected the climate conditions and we have made this more explicit in the text.

  • Line 166: Need more details here. I believe you used the mean soil moisture and temperature. You should use a MIXED model ANOVA. What about the Year effects? Did you test the interaction of climate treatment and cropping systems. See my previous comment.

There was a year effect, and these were the models we utilized. You can see one is with climate treatment and one is without. We also included an autocorrelation structure. Times is day of the year.

gmodel  <- gam(feclimm$Value.mean ~ feclimm$year + feclimm$ClimTreat + s(feclimm$Time, by= feclimm$ClimTreat), correlation = corARMA(1|Time))

gmodel1 <- gam(feclimm$Value.mean ~ feclimm$year + s(feclimm$Time), correlation = corARMA(1|Time))

anova (gmodel1, gmodel, test='F')

We clearly recognize there is a year effect, and we discuss it in the manuscript, and cite, that climate manipulations are dependent on the ambient conditions. We were most focused on intra year contrasts, and the response of weed biomass and diversity.

  • Lines 166-167 I do not see this ANOVA results in the result section 

The ANOVA results can be found on the revised version of Results section.

  • Line 168: generalized or general? if you used generalized you have to provide the distribution you used. At the same time you mention about log transformation which does not make sense. 

They were linear mixed effects models and we have removed “generalized” from the text. Thank you for pointing out this discrepancy.

  • Line 172 what you mean by year in tis study? Did you repeat the data collection then how? Was it a fully-phased design? you have mention this clearly at the beginning 

As explained in the first line of the Material and Methods section, this study was conducted during the 2015-2016 and 2016-2017 growing seasons.  Thus, Reviewer 3 is correct, we collected data during two years.  In the revised version of this manuscript, we explain that this was a fully phased study (i.e., all crop phases were present every year).

  • Line 174 I believe all the figures are in backtransformed scale. 

Yes, Figure 1 is back transformed and we have clarified this in the figure legend. Thank you for pointing out this omission.

  • Line 178 again if you are using generalized models you need to give the details.

We have clarified that Simpson’s was calculated using weed biomass and we give fixed effects and random effects for both the response variables (richness and Simpson’s).

  • Line 194: In all the three cropping systems? Did you ever tested that there is a cropping system by climate treatment interaction? I think this is important as we do not know how the soils in different cropping system response to changes in ambient climatic conditions. Isnt this will be one of your main hypothesis.

Please see our response above

  • Line 213: per square meter? better indicate all the values in kg/ha which is the standard.

For weed biomass, we have scaled our results to g m-2. However, we believe that because weeds are patchy it is not representative to scale weed biomass up to kg ha-1.

  • Table 1 (line 220) I would suggest to include all the weed variables in this table including richness simpson index. Remove unnecessary information such as sums of squares and means squares, f values.  PLEASE SEE QUESTIONS IN THE PDF FILE – Line 222

Given that they are different models (lmer v. glmer) we chose to keep Table 1 as it is, however, we have re-written the richness and Simpson’s paragraphs for better clarity.

  • I do not get how you get 2 years data in this rotation. Unless it is a fully phased design?  

As explained above, this study was conducted study during the 2015-2016 and 2016-2017 growing seasons.  In the revised version of this manuscript, we explain that all crop phases were present each year.

  • Figure 1. better to change the y axis scale to g/m2 or kg/ha

As explained above, for weed biomass we have scaled Figure 1 to g m -2, however because weeds are patchy it is not representative to scale weed biomass up to kg ha-1.

  • According to Table 1 there is no temperature x moisture x cropping systems x year interaction? So you do not  need to show this in figure 1? What you have to show in a figure is cropping system x Year interaction

We have changed Figure 1 to represent just the significant factors (year and cropping system). Thank you for the suggestion.

  • Line 224: here you say total weed biomass? The y axis says total weed biomass of weed species. Better remove species there

Thank you we have changed the y axis label.

  • lines 237-242 It is better to show all these info first in ANOVA table. It is important to follow the the effects in hierarchical way

Given that they are different models (lmer v. glmer) we chose to keep Table 1 as it is. However, we have re-written the richness and Simpson’s paragraphs for better clarity.

  • Figure 2: line 246 Is this for both years?  Were these calculated using biomass?

We went back and re-ran our analyses and discerned a yearly effect, therefore we have included year in both the richness and Simpon’s results and graphics (Figs. 2 and 3). Simpson’s diversity index was calculated using species biomass and we have added a sentence in the methods section clarifying this.

  • Figure 3 Why dont you ovrelay the main species to this ordination diagram? I know it is possible in most ordinations

We considered it but did not include this information because it would make a very busy and unintelligible graphic. Also, as explained above, cropping system is the most important factor, which is also covered in Larson et al 2021.

  • Figure S2  change the Y axis title to Mean soil temperature

Modified as suggested.

  • Fig S3  variation or actual soil moisture? Better to use changes in daily soil moisture

Soil moisture. Thank you for pointing out this discrepancy.

Discussion

  • It is not properly structured and organized. Start with the main interactions observed/not observed and discuss around that. Since the climate is a major factor under control in this study, start the discussion about the effect of climate. The discussion needs to be strengthened with further discussion of observations. You discuss species shift, but these data were shown in supplementary tables, not in the main text.

As indicated by Reviewer 3 climate is “major factor under control in this study.” However, our goal was to assess the joint impact that cropping systems and predicted temperature and moisture conditions on weed communities. As highlighted in our manuscript, while cropping system had the major impacts on weed communities, the impact of temperature and moisture manipulations was more subtle.  Thus, we believe it will be misleading to emphasize the role that climate had on the structuring of weed communities over the importance of cropping systems. Still, we have carefully revised the Discussion section to (1) emphasize the pressures that unprecedent climatic conditions create on the sustainability of cropping systems, (2) indicate that weed community responses to to temperature and moisture manipulations differed across variables, and (3) clarify that the short-term nature of our study couple with the relatively small size of the Open Top Chambers and Rain Out Shelters we utilized “could have masked our ability to detect the impacts of predicted warmer and drier climate conditions on weed communities across contrasting cropping systems” We have also modified the Abstract to better reflect the impact  of climate on weed communities.

  • Conclusion- Needs to focus on what you found and what not.

As suggested by Reviewer 3, in the revised Conclusion, we highlight the major results and implications of our research.  Specifically, we (1) indicate that “while cropping system characteristics represents a major driver structuring weed communities, short-term impact of changes in temperature and moisture conditions appear to be more subtle”, (2) summarize the major weed community differences across cropping systems and indicate potential ecological implications of the observed results, and (3) emphasize the need to assess “the mechanisms underpinning the assembly of weed communities under current and predicted climate scenarios to integrate ecologically-bases processes”, a required step in the designing of sustainable cropping systems.

Round 2

Reviewer 3 Report

The authors have done a fair amount of work on revising the manuscript.  I still find that the results section can be improved if the suggested changes were followed. Particularly, the results in the S3 table should be moved to the body of the manuscript as this information is highly used throughout the manuscript. 

The title might need some modifications as the discussion about winter cereal is almost does not exist thus highlighting it in the title does not make sense

Axis titles in theY axis need to be improved particularly with reference to the scales of measurments.

Some of the important and major conclusions are not included in the abstract (See the comments attached).
